

# *Burkholderia pseudomallei* type III secreted protein BipC: role in actin modulation and translocation activities required for the bacterial intracellular lifecycle

Wen Tyng Kang[1], Kumutha Malar Vellasamy[1], Lakshminarayanan Rajamani[2], Roger W. Beuerman[2] and Jamuna Vadivelu[1]

[1] Department of Medical Microbiology, Faculty of Medicine, University of Malaya, Kuala Lumpur, Malaysia
[2] Antimicrobials, Singapore Eye Research Institute (SERI), Singapore, Singapore

## ABSTRACT

Melioidosis, an infection caused by the facultative intracellular pathogen *Burkholderia pseudomallei*, has been classified as an emerging disease with the number of patients steadily increasing at an alarming rate. *B. pseudomallei* possess various virulence determinants that allow them to invade the host and evade the host immune response, such as the type III secretion systems (TTSS). The products of this specialized secretion system are particularly important for the *B. pseudomallei* infection. Lacking in one or more components of the TTSS demonstrated different degrees of defects in the intracellular lifecycle of *B. pseudomallei*. Further understanding the functional roles of proteins involved in *B. pseudomallei* TTSS will enable us to dissect the enigma of *B. pseudomallei*-host cell interaction. In this study, BipC (a translocator), which was previously reported to be involved in the pathogenesis of *B. pseudomallei*, was further characterized using the bioinformatics and molecular approaches. The *bipC* gene, coding for a putative invasive protein, was first PCR amplified from *B. pseudomallei* K96243 genomic DNA and cloned into an expression vector for overexpression in *Escherichia coli*. The soluble protein was subsequently purified and assayed for actin polymerization and depolymerization. BipC was verified to subvert the host actin dynamics as demonstrated by the capability to polymerize actin *in vitro*. Homology modeling was also attempted to predict the structure of BipC. Overall, our findings identified that the protein encoded by the *bipC* gene plays a role as an effector involved in the actin binding activity to facilitate internalization of *B. pseudomallei* into the host cells.

# INTRODUCTION

Melioidosis, a potentially fatal disease, is a neglected tropical disease that afflicts both humans and animals. It is caused by the Gram-negative soil saprophyte, *Burkholderia pseudomallei*. It has varied clinical presentations, including asymptomatic infection, chronic

Corresponding author
Jamuna Vadivelu,
jamuna@ummc.edu.my,
jamuna@um.edu.my

pneumonia, and fulminant septic shock with abscesses in multiple internal organs (*Wiersinga, Currie & Peacock, 2012*). This disease is endemic across parts of tropical South East Asia and Northern Australia (*Sun & Gan, 2010*). Most cases of melioidosis are the result of percutaneous inoculation following exposure to the bacteria from muddy soils or surface water (*Chaowagul et al., 1989*). The etiological agent has been classified as Tier 1 Select Agent by the United States Centers for Disease Control and Prevention (CDC) due to its high mortality rate that could be used as a potential agent for bioterrorism (*Sarovich et al., 2014*).

Melioidosis was classified as an emerging disease with a steady increase in the number of patients over the past few years (*Dance, 2002*). In addition, the fulminating septicemia form of melioidosis typically has a mortality rate of greater than 90%. Previous studies have demonstrated that *B. pseudomallei* is able to survive and replicate in both the phagocytic and non-phagocytic cells (*Stevens et al., 2004*). However, the most striking feature of this bacterium is the ability to remain latent in the host for up to 62 years and cause recrudescent infections following many years past the initial infection (*Ngauy et al., 2005*). Among the virulence factors of *B. pseudomallei*, the long dormancy state may be attributed to the type III secretion system (TTSS), which facilitate the pathogen to survive and replicate in both phagocytic and non-phagocytic cell (*Brett & Woods, 2000*; *Stevens et al., 2002*). However, the exact mechanism of this phenomenon is still yet to be discovered.

TTSS involves a cluster of genes encoding a series of proteins that has been reported to play a role in the pathogenicity of many Gram-negative bacterial pathogens (*Mecsas & Strauss, 1996*). These pathogenic bacteria use the TTSS to deliver virulence factors, also known as effector proteins, from the bacterial cytoplasm into the host cell interior. The effector proteins function to facilitate entry into and survival of the bacteria in the phagosome of the host cells (*He, Nomura & Whittam, 2004*). Recent reports have indicated that one of the protein products of TTSS3, BipC, was present in the secretome of *B. pseudomallei* laboratory culture and it was found to be immunogenic as verified by reactivity to mice anti-*B. pseudomallei* sera (*Vellasamy et al., 2010*). This protein has been previously postulated as a protein translocator (*Stevens et al., 2004*; *Sun & Gan, 2010*). Based on our previous finding, mutation in the *bipC* gene has been shown to impair the ability of *B. pseudomallei* to adhere, invade, and survive intracellularly in the epithelial cells, and also involved in the actin-tail formation. In addition, BipC is also required for full virulence in a murine model of melioidosis and cytotoxicity *in vitro* (*Kang et al., 2015*).

These findings demonstrated that BipC plays a role in *B. pseudomallei* pathogenesis, however, there are many questions that exist as to how BipC functions as potential effector protein and promote cell invasion. Therefore, it is of great interest to elucidate the structure and properties of BipC in order to obtain more information about the exact roles played by this protein. This present study that combines the *in silico* and *in vitro* studies were performed to understand the structure and possible functions of the BipC protein. These efforts are important in establishing the biological role and importance of BipC in the virulence of *B. pseudomallei*.

## MATERIALS AND METHODS

### Bacterial strains, plasmids, and culture conditions

The bacterial strains and plasmids used in this study are listed in Table S1. *B. pseudomallei* K96243 (*Holden et al., 2004*) was cultured and maintained in Luria-Bertani (LB; Difco, Detroit, Michigan, USA) broth. All plasmids were propagated in *E. coli* Top10. The *E. coli* BL21 (DE3) was used for cloning and expression purposes. pET-30a(+) (Novagen; EMD Biosciences, Germany) was used as an expression vector to express the cloned *bipC* gene. Unless stated specifically otherwise, all bacterial strains used were grown at 37 °C on LB agar and broth (Difco, Detroit, Michigan, USA) containing appropriate antibiotics.

### Sequence analysis, homology modeling, and model assessment

The linear chain of BipC containing 419 residues was submitted to various sequence analysis on SWISS-PROT (*Bairoch & Boeckmann, 1992*), Basic Local Alignment Search tool (BLAST) (*Altschul et al., 1990*) and Protein Data Bank (PDB) (*Berman et al., 2000*). Type three secretion effector (TTSE) translocation signal peptide online program, such as Mod-Lab (http://gecco.org.chemie.uni-frankfurt.de/T3SS_prediction/T3SS_prediction.html) (*Lower & Schneider, 2009*), T3SEdb (http://effectors.bic.nus.edu.sg/T3SEdb/predict.php) (*Tay et al., 2010*) and Effective T3 (http://www.effectors.org/index.jsp) (*Arnold et al., 2009*) were used to predict the presence of TTSE for BipC. The domain family analysis was performed using Pfam protein families database (*Finn et al., 2010*). Multiple sequence alignment between BipC and the templates were performed using CLUSTALW (*Thompson, Higgins & Gibson, 1994*). The information regarding the secondary structure of BipC was obtained from the online bioinformatics tool which known as GOR4 secondary structure prediction (http://npsa-pbil.ibcp.fr/cgi-bin/secpred_gor4.pl) (*Riedel et al., 2006*). Prediction of intrinsically unstructured regions of BipC was performed using PONDR (http://www.pondr.com/) (*Li et al., 1999*). The template for structure prediction was chosen based on the result from the pDomThreader (*Lobley, Sadowski & Jones, 2009*), (PS)$^2$-v2 (*Chen, Hwang & Yang, 2009*), and RaptorX (*Kelley et al., 2015*) analyses. Verification of the built model was done using PROCHECK (*Laskowski et al., 1993*).

### Cloning and expression of BipC protein

*B. pseudomallei* K96243 was grown overnight at 37 °C in LB broth. DNA was extracted using Wizard® genomic DNA purification kit (Promega, Madison, Wisconsin, USA) according to the manufacturer's instruction. The 1,260 bp fragment of the *bipC* was amplified by PCR using a pair of specific designed primers. The primers used were as follows: forward primer 5′-CCCAAAGGATCCACGAAGTCCAAGAGGTGCGT-3′ and reverse primer 5′-CCCAAAAAGCTTTCAGGTCCGCAGATTGCC-3′. *BamH*I and *EcoR*I site is underlined in the forward and reverse primer, respectively. The PCR reaction mixture (50 μl) contained 500 ng DNA, 0.2 mM dNTP, 1× *Taq* buffer with KCl, 1.0 mM MgCl$_2$, 0.2 μM of primers, and 0.25U *Taq* DNA polymerase (Invitrogen, Carlsbad, California, USA). The PCR was performed for a cycle of denaturation at 95 °C for five minutes, followed by 30 cycles at 95 °C for one minute, annealing at 62 °C for one minute and extension at 72 °C for one minute,

and lastly further extension at 72 °C for five minutes. Subsequently, the amplicon was purified using the QIAquick PCR Purification Kit (Qiagen, Venlo, Netherlands).

The obtained PCR fragments were cloned into pCR®2.1-TOPO (Promega, Madison, Wisconsin, USA) and subjected to sequencing (Macrogen, South Korea). The confirmed *bipC* fragment was then cloned into *BamH*I and *EcoR*I sites of pET30a(+). For protein expression, the recombinant plasmid, pET30a::*bipC* was transformed into *E. coli* strain BL21(DE3) and the transformants were selected on LB agar containing 50 μg/ml kanamycin. A transformant was chosen and cultured overnight at 37 °C in LB broth containing 50 ug/ml kanamycin. Subsequently, the expression of histidine tagged BipC was induced using 1.0 mM isopropyl thiogalactoside (IPTG). Following the induction period, cells were pelleted and re-suspended in 5 mL pre-cooled buffer (10 mM imidazole, 50 mM Tris-HCI, pH 7.2 and 300 mM sodium chloride). The cells were then sonicated and centrifuged at 4,000 × g for 10 min.

## Purification of BipC protein

The expressed protein from the cell culture supernatants was first purified by Nickel-nitrilotriacetic acid (Ni-NTA) affinity chromatography using QIAexpress® Ni-NTA fast start kit (Qiagen, Venlo, Netherlands) according to the manufacturer's instruction. Briefly, 250 ml cultures of *E. coli* cells were disrupted by lysis with 10 ml native lysis buffer and incubated on ice for 30 min. The cell lysate was centrifuged at 4 °C to pellet the cellular debris and the resulting cell supernatant was then transferred to a fast start column. The flow-through fraction was collected. Next, the column was washed two times in 4 ml of native wash and eluted using native elution buffer. The first purified protein was further purified using size exclusion chromatography (SEC). In brief, 100 μl of BipC protein was added to a Sephadex® size exclusion column (Amersham Biosciences, Buckinghamshire, UK) with 1 ml increments, at a rate of 1 ml/min, for 30 min. The sample was allowed to settle into the bed of the column and eluted with 10 mM phosphate-buffered saline (PBS). Aliquots were collected at one minute intervals and ultraviolet (UV) absorbance spectra of the eluents were analyzed. The purity of the protein preparation was finally assessed by sodium dodecyl sulphate polyacrylamide gel electrophoresis (SDS-PAGE). Following reaction with size exclusion filtration as described above, the BipC fractions eluted were analyzed with high performance liquid chromatography (HPLC). Total soluble protein concentrations were then determined by the Bradford protein assay (*Bradford, 1976*).

## Far UV-Circular dichroism (CD) spectropolarimetry of BipC

BipC protein was prepared to a concentration of 0.05, 0.10, 0.25, and 0.50 mg/ml in 10 mM phosphate buffer (pH 7.0). CD spectropolarimeter (Jasco J-810, Great Dunmow, Essex, UK) was purged with nitrogen gas for 15 min prior to introduction of sample. The CD spectrum was recorded in phosphate buffer solution (PBS) in the far UV region (190–260 nm) and all the scans were background subtracted with the scans of buffer alone. Thermal denaturation was performed with different temperature/wavelength (20–90 °C/222 nm) to infer the thermal stability of the protein. The measurements were performed using a 0.1 cm path-length cuvette at 25 °C. Four scans were recorded for each spectrum. An average

result was taken and baseline subtracted. The mean residual weight ellipticity ($\theta_{MRW}$) was calculated using the following equation:

$$[\theta]_{mrw} = ([\theta]_{obs} \times MRW)/(10 \times c \times l)$$

where $[\theta]_{obs}$ is the observed ellipticity in millidegrees, MRW is molecular weight of protein divided by the number of peptide bonds, $c$ is the protein concentration and $l$ is the optical path-length of the cuvette.

## Cytotoxicity assay of BipC

To determine the cytotoxicity of BipC, human lung epithelial cell line A459 (ATCC® CCL-185™) was first seeded at $5 \times 10^4$ cells/well in a 96-well plate. Aliquots of BipC protein with final concentrations of 0.05-1 mg/ml in the plate were added to the cells and incubated overnight at 37 °C with 5% $CO_2$. The released cytolytic lactate dehydrogenase (LDH) was determined using the CytoTox96 kit (Promega, Madison, Wisconsin, USA) according to the manufacturer's instructions. In brief, a 100 µl aliquot of the supernatant obtained from each well was added to a 96-well plate. The LDH release (% cytotoxicity) was then calculated using the following equation: ($OD_{490}$ experimental release − $OD_{490}$ spontaneous release)/($OD_{490}$ maximum release − $OD_{409}$ spontaneous release) × 100. The spontaneous release was the amount of LDH released from the cytoplasm of uninfected cells, whereas the maximum release was the amount released by total lysis of uninfected cells using Triton X-100. The assays were performed in triplicate and repeated twice. The phenotype of A549 cells following 24 h of post-exposure to the different concentrations of BipC was observed under the microscope. The uninfected cells served as negative control.

## Actin polymerization and depolymerization assays

Actin polymerization assay was performed using Actin polymerization biochem Kit™ (Cytoskeleton, Inc., *Denver, Colorado,* USA) according to the manufacturer's instructions. In brief, stock of purified His-BipC protein (1 mg/ml) was prepared in an actin compatible buffer. For polymerization, G-actin stock (0.4 mg/ml) was prepared with G-buffer. The mixture was then incubated on ice for one hour in order to depolymerize actin oligomers that have formed during storage. Pyrene actin and unlabeled actin were mixed 1:1 and diluted to a final concentration of 200 µg/ml in G-buffer. The mixture (200 µl) was then transferred into 96-well microtiter plates. Baseline fluorescence was monitored for five minutes at 37 °C using a Fluoroscan fluorescence microtiter plate reader (Labsystems, Helsinki, Finland) with $\lambda_{excitation}$ at 355 nm and $\lambda_{emission}$ at 405 nm.

Polymerization was initiated by the addition of 10× actin polymerization buffer followed by mixing for 10 s. Actin polymerization, as determined by an increase in pyrene fluorescence, was monitored at various time points. For treatment with BipC, serial dilutions of BipC were prepared in $ddH_2O$ from 175 mM stock and 20 µl was added per ml pyrene actin mixture. The samples were examined over a period of one hour immediately after adding BipC protein. Controls included a similar volume of buffer (vehicle control). For depolymerization, stock of pyrene F-actin (1 mg/ml) was prepared by adding G-buffer. The G-actin was polymerized to F-actin by adding 10 µl of 10× actin polymerization

**Table 1** Summary of TTSE signals peptide prediction results to be present in the sequence of BipC.

| Program | Cut off score | Score | Prediction |
|---|---|---|---|
| ModLab | 0.400 | 1.05000 | Yes |
| T3SEdb | ≈ 1.000 | 1.00000 | Yes |
| Effective T3 | 0.999 | 0.52961 | Yes |

buffer and incubated for one hour at room temperature. F-actin samples (200 µl/well) were aliquoted into microtiter plates, and the pyrene fluorescence was monitored for one hour. For BipC-treatment, 20 µl BipC (0–250 µg/ml) was added to each well before analysis.

### Protein pull down assay

Protein–protein interaction of BipC was performed using Dynabeads® His-tag isolation and pull-down kit (Novagen, EMD Biosciences, Germany) according to the manufacturer's instructions. Sample of the histidine-tagged BipC, G-actin, and F-actin was prepared, 50 µl of Dynabeads® mixed with His-BipC and the mixture was incubated on a roller for five minutes at room temperature. The tube was placed on the magnet for two minutes and then the supernatant was discarded. The beads were washed four times with 300 µl wash buffer by placing the tube on a magnet for two minutes and the supernatant was discarded. G-actin or F-actin was then added to the bead/BipC complex. The mixed sample was incubated on a roller for 30 min at room temperature, beads washed four times with 300 µl of wash buffer by placing the tube on a magnet for two minutes and the supernatant was discarded. To elute bound proteins, 100 µl his-elution buffer was added and the suspension was incubated on a roller for five minutes at room temperature. The beads at the tube wall were collected using a magnet and the supernatant containing the His-BipC and its interacting monomeric G-actin or filamentous F-actin protein was transferred to a clean tube.

## RESULTS

### BipC sequence analysis

The bioinformatics tools (ModLab, T3Sedb, and Effective T3) used in this study predicted BipC as a TTSE, which is secreted directly into the cytosol of the host cell, based on the translocation signal peptide that was predicted to be present in the sequence of BipC (Table 1). The sequence analysis with a simple BLAST search against non-redundant (NR) database and Pfam software showed that BipC harbor a conserved domain of IpaC-SipC superfamily. Thus, the web server ClustalW was used for multiple alignments analysis of BipC, SipC (*Salmonella*), and IpaC (*Shigella*). In the sequence alignment, BipC showed that some of the residues were strictly conserved with SipC and IpaC (Fig. 1). ClustalW results showed a similarity between the target and the templates. SipC and IpaC harbors actin binding sequences at the C-terminal region. Thus, similar presence of highly conserved residues at the C-terminal of BipC may indicate that this protein also comprise of the actin binding sequences. This domain present at the C-terminal region was also predicted to form the alpha helix secondary structure which is required for actin binding.

The GOR secondary structure prediction tool predicted BipC to contain approximately 70% of alpha helical conformations and 22% of random coil conformations (Fig. S1). The

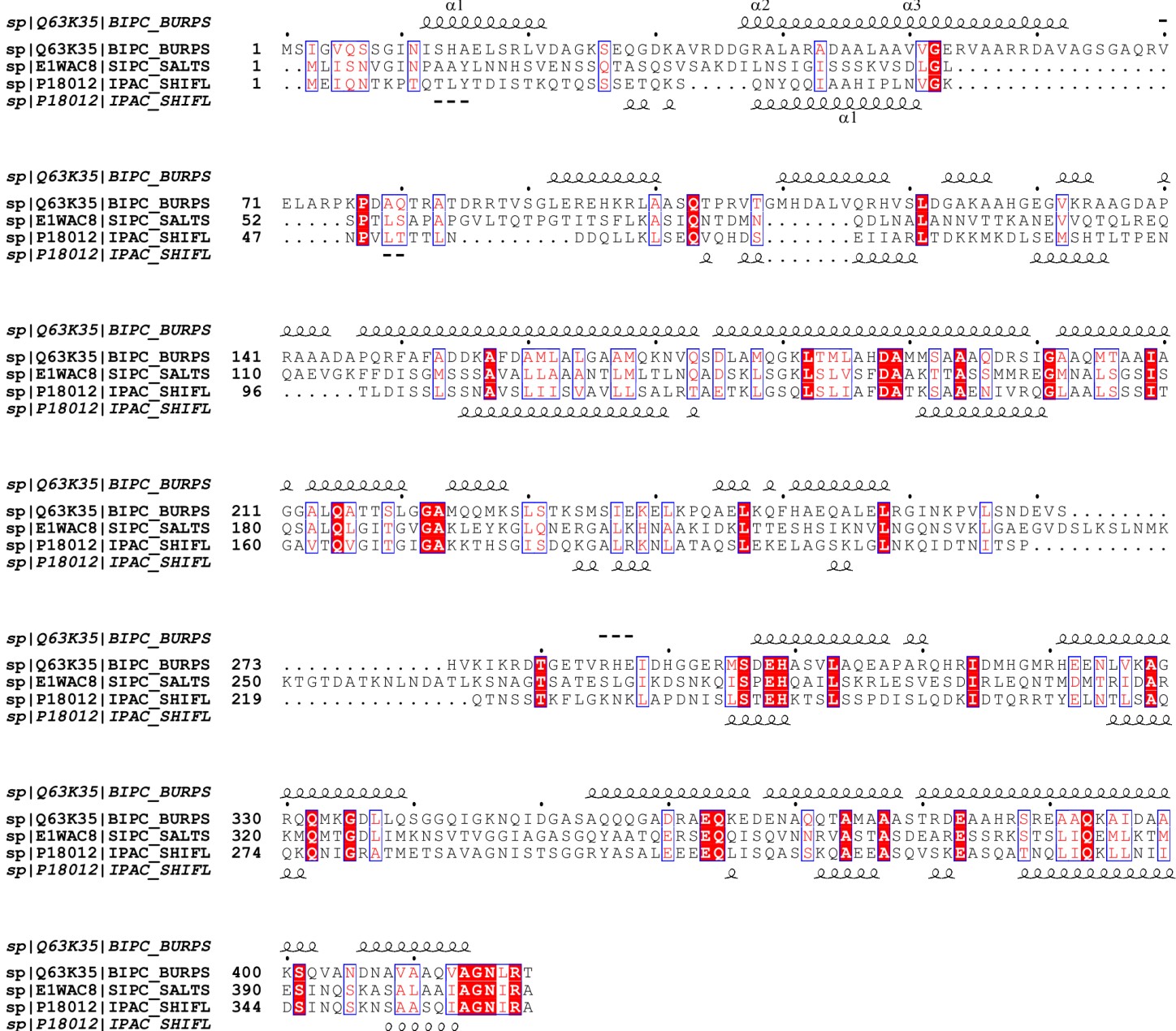

**Figure 1** **Alignment of the sequences of BipC protein from *B. pseudomallei* with the SipC protein from *Samonella* and IpaC protein from *Shigella*.** The secondary structure of BipC is shown on top of the aligned sequences. Positions for which the percentage of 'equivalent' residues, considering their physico-chemical properties, is higher than 70% are colored in red on a white background. Strictly conserved residues are colored in white characters on a red background.

folded $\alpha$-helix region may be crucial in the functionality of BipC. Prediction of the intrinsically unstructured regions of BipC was performed using the online program, PONDR. Using this software, the scores of higher than 0.5 are considered as disordered region and scores that are lower than 0.5 is considered as ordered. The results predicted an amphipathic structure comprising of alternative ordered and disordered regions for BipC (Fig. S2).

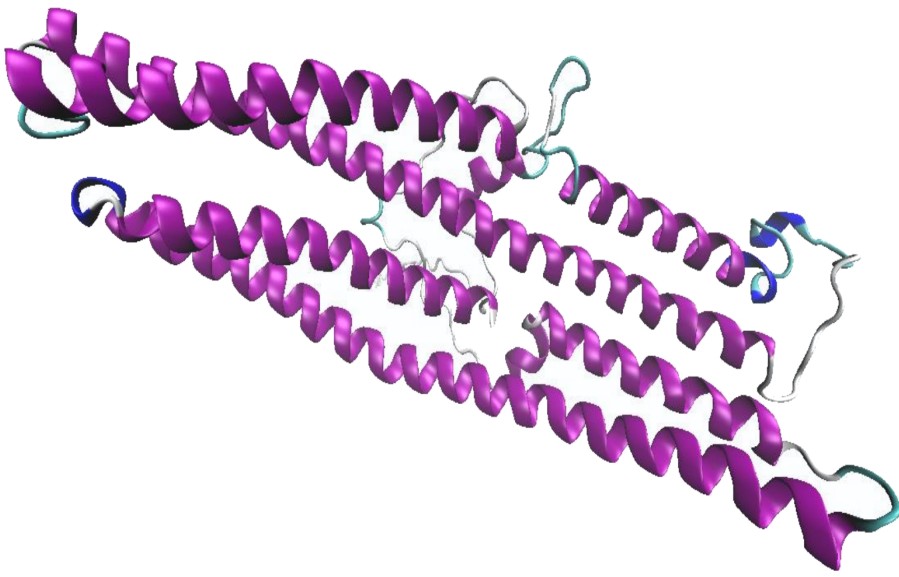

**Figure 2** **The predicted BipC model using 1wp1B01 as a template for homology modeling with the best scoring from Ramachandran plot.** Alpha helix secondary structures are represented in purple. The graphic was generated using Visual Molecular Dynamics visualization tool.

## Template-based homology modeling

BipC amino acid residues were subjected to sequence analysis on BLAST search with Protein Data Bank (PDB) for a potential template for homology modeling. The results yield did not show any good E-value and only three structures were identified with the hits below E-value threshold. All the hits obtained shared low sequence identity with only approximately 30% within a small coverage in the sequence. Since there were no potential structural template, the web servers' pDomThreader, (PS)$^2$-v2, and RaptorX were therefore used for protein fold recognition in order to further identify the potential template for structural modeling.

From pDomThreader analysis, 1wp1B01 was identified as the best template with a high confidence level. 1wp1B01 is an X-ray structure of the outer membrane protein from *Pseudomonas aeruginosa* with the length of 405 amino acid residues and was selected as the template for model building. On the other hand, (PS)$^2$-v2 analysis identified 1tr2B as the best template with the coverage of 92.6% of the whole sequence length. 1tr2B is an X-ray structure of a 1,066 amino acid human full-length vinculin. However, the best template selected from RaptorX is 3dyjA. This structure is a 332 amino acid talin-1 from mus musculus. These templates obtained from both (PS)$^2$-v2 and RaptorX indicated that BipC shared structural features with actin binding domain, whereas template obtained from pDomThreader determined that BipC shared structural features with the transporter domain. Three of the 1wp1B01, 1tr2B, and 3dyjA were selected as the template for model building. The built models were then validated and the result indicated that ~99% of the total residues fell within the most favorable and additional allowed regions for the model built using 1wp1B01 template (Fig. S3). In the 3 dimensional (3D) homology model of BipC, abundant alpha helix regions were displayed (Fig. 2). Coupled with the Ramachandran plot, this model can be accepted as the best potential model representing the 3D structure of BipC protein.

## Cloning, expression, and purification of BipC

A 1,260 bp open reading frame (ORF) of *bipC* gene was amplified from *B. pseudomallei* K96243 genomic DNA by PCR and it was cloned into a pET30a(+) vector for the expression of BipC as a 6× histidine tagged (His$_6$-tagged) protein in *E. coli*. Approximately 90% of the His$_6$-tagged BipC was present as soluble fraction following the Ni-NTA purification. The Ni-NTA purified protein was further purified using Sephadex® size exclusion column. Here is shown the first effort to purify the protein sample in order to determine the structure of BipC (Fig. 3). Size exclusion successfully removed the excess molecules from the Ni-NTA purified BipC. This method of purification allowed us to obtain approximately 95% pure protein through High performance liquid chromatography (HPLC) analysis. Float-A-Lyzer dialysis device was then used for desalting of the fraction collected prior to CD analysis.

## Circular dichroism (CD) of purified BipC

Far-UV CD spectrum was observed for BipC to determine the secondary structure of the protein. The appearance of intense negative minimum around 207 and 222 nm as well as a positive maximum around 195 nm confirmed the existence of a dominant $\alpha$-helical structure (Fig. 4A). In order to determine the thermal stability, viable temperature CD was recorded for BipC. The temperature scans indicated the absence of a sharp transition even above 90 °C, indicating excellent thermal stability of the protein under physiological conditions (Fig. 4B).

## Cytotoxicity of BipC protein

In order to identify the cytotoxic effect of BipC, A549 cells were incubated with purified His-tagged BipC. A concentration-dependent increase in the LDH activity was observed after 24 h of exposure. At the lowest concentration tested (5 μg/ml), about 4.5% of the LDH release was observed. Release of LDH was found to increase proportionally with the increase in the concentration of BipC protein exposed to the A549 cells. The highest level of LDH release (51%) was observed upon exposure to 1 mg/ml of the purified BipC protein. The induction of cell death was observed as compared with the untreated cells (Fig. 5A) and this data indicated that the BipC may trigger cell death to the human lung epithelial cell lines.

In addition, the phenotype of the A549 cells exposed to the different concentration of purified His-BipC was observed using inverted microscopy (Fig. 5B). The control cells (without exposure to BipC) demonstrated approximately 90% high confluency (Fig. 5B, panel i). However, the A549 cells that were exposed to BipC demonstrated decreased confluency with the increasing concentration of BipC. At the higher concentration of BipC (500–1,000 μg/ml), the cells gradually became irregular, shrunken, and detached from the cell culture substratum (Fig. 5B, panels ii, iii and iv). These changes were characteristic of cell death.

## Actin polymerization and depolymerization of purified BipC *in vitro*

Pyrene-actin polymerization assay was performed in order to determine the involvement of BipC in the assembly of actin filaments. In this assay, fluorescence of pyrene-actin increased significantly when G-actin monomers were incorporated into a filament, permitting polymerization to be measured in real-time. His-BipC stimulated actin polymerization in the reactions containing 2 μM actin (5% pyrene-labeled) with a dose-dependent manner

**(A)**

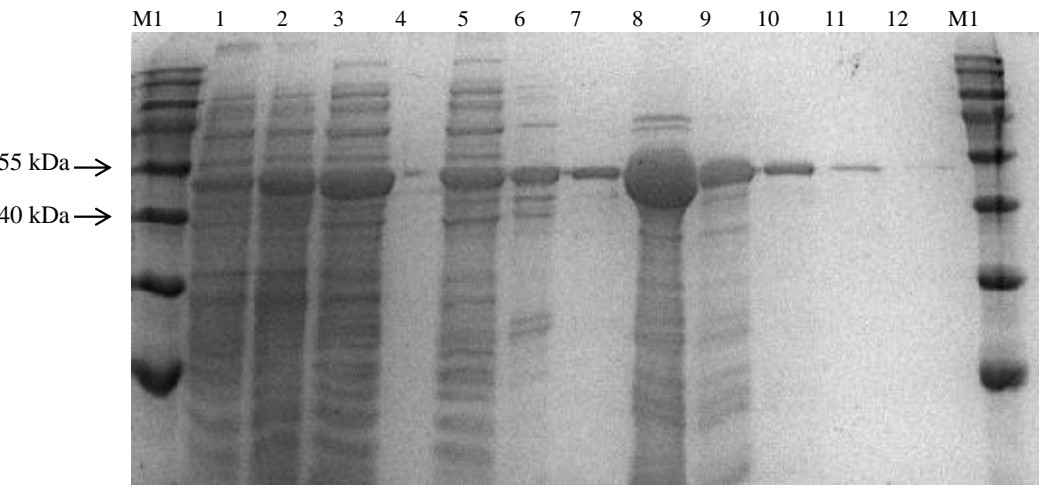

**(B)**

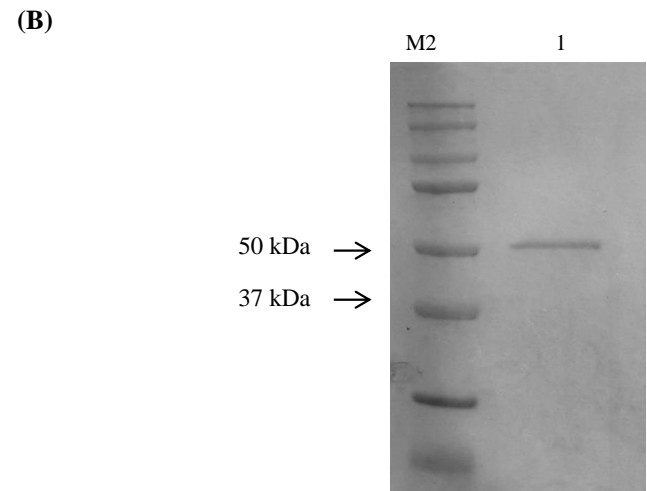

**Figure 3 Expression and purification of BipC.** (A) Purification of fusion protein His-BipC using Ni-NTA affinity chromatography. Lane M1, PageRuler prestained protein ladder; lane 1, uninduced pET30a(+)::*bipC*; lane 2, induced pET30a(+)::*bipC*; lane 3, cell lysate of induced pET30a(+)::*bipC*; lane 4, storage buffer; lane5, protein flow-through; lane 6–7, wash samples; lane 8–12, elution of purified His-BipC protein on 12% SDS-PAGE. (B) Size exclusion chromatography purification of His-BipC protein. Lane M2, Precision Plus protein ladder; lane 1, purified BipC protein.

(Fig. 6A). Saturation occurred at approximately 62.5 ug/ml of BipC and this concentration showed about 3-fold increase for the actin polymerization. The data obtained with this *in vitro* actin polymerization assays demonstrated that BipC has the ability to enhance actin polymerization without the requirement of additional proteins.

Besides that, the effect of BipC on F-actin depolymerization kinetics was also monitored in this study. In the presence of BipC, the percentage of pyrene-labeled F-actin was constant

**(A)**

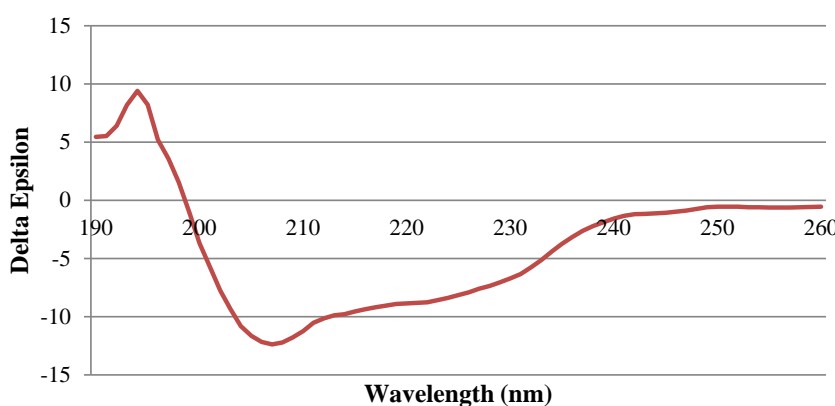

**(B)**

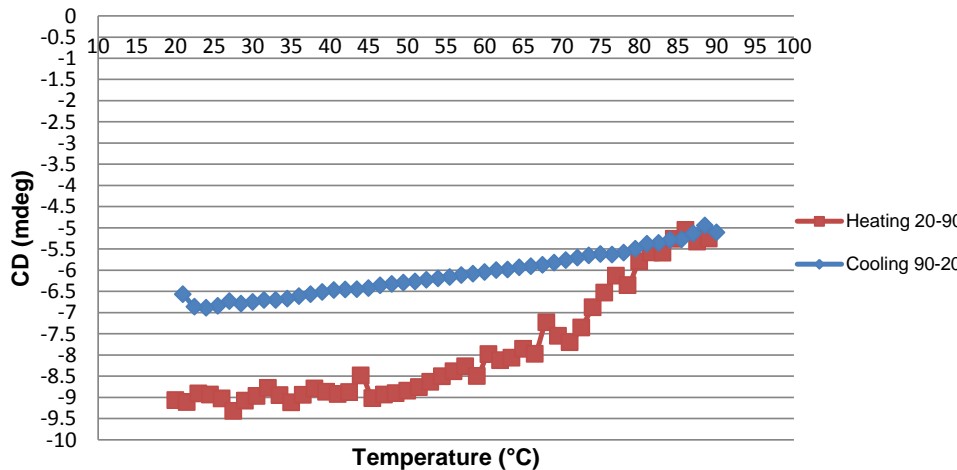

**Figure 4  Circular dichroism spectropolarimetry of the His-BipC protein.** (A) Circular dichroism spectra of BipC in 10 mM PB, pH 7.2 at 25 °C. (B) Changes of BipC protein complexes as a function of temperature used to determine the thermodynamics of folding. The viable temperature of BipC confirming the preponderance of random coil confirmation in the structure. The protein was diluted to a concentration of 0.50 mg/ml in 10 mM phosphate buffer (pH 7.2) to record the spectrum.

over 60 min, in comparison with the control (only F-actin) where only 30% remained (Fig. 6B). BipC inhibited the rate of F-actin depolymerization and hence, greatly increased the stability of F-actin. F-actin stabilization by higher concentration of BipC was greater than that induced by the lower concentration. This protein was shown to have the ability to repolymerize actin that was depolymerized so that a steady level of polymerized actin was detected. Taken together, these results suggest that BipC was able to bind F-actin and modulated actin dynamics *in vitro*.

**(A)**

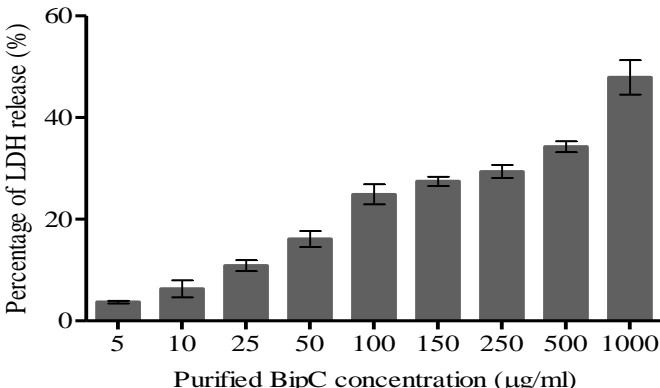

**(B)**

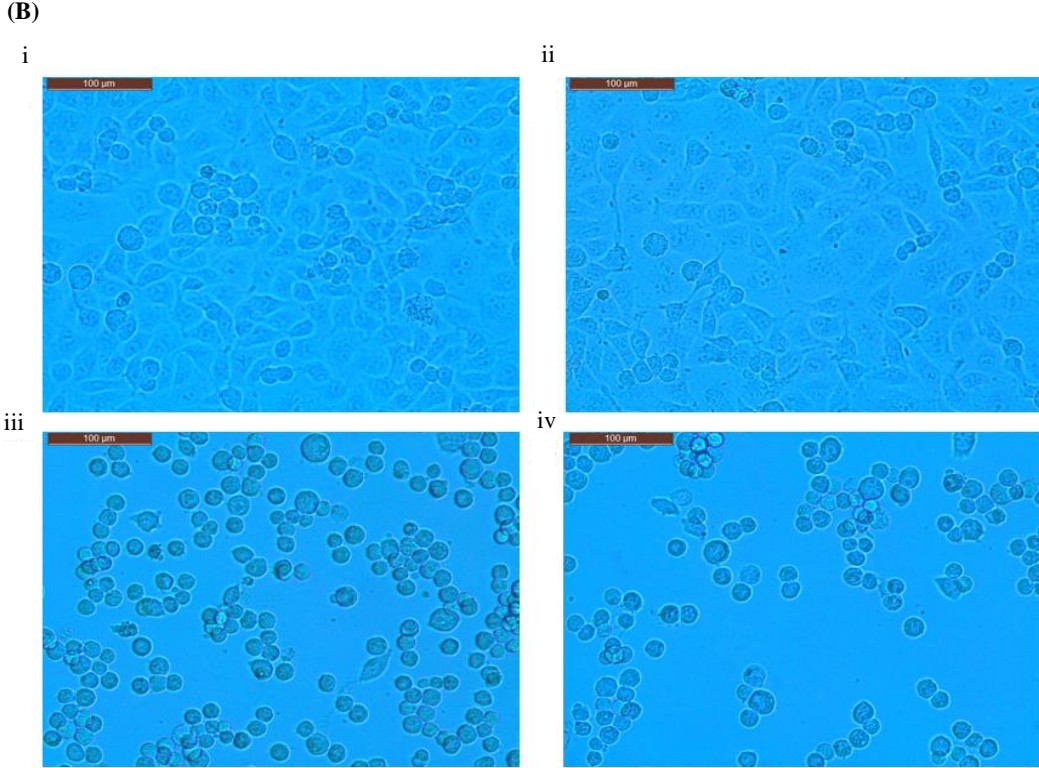

**Figure 5  Cytotoxicity of BipC.** (A) Percentage of Lactate dehydrogenase (LDH) released was assayed in order to determine the cytotoxicty in A549 human lung epithelial cells exposed to BipC. The percentage of LDH released is directly proportional to the concentration of BipC. (B) A549 epithelial cells exposed to (ii) 50 $\mu$g/ml, (iii) 500 $\mu$g/ml, (iv) 1,000 $\mu$g/ml of purified BipC, and (i) untreated cells (control) viewed under the light microscope.

**(A)**

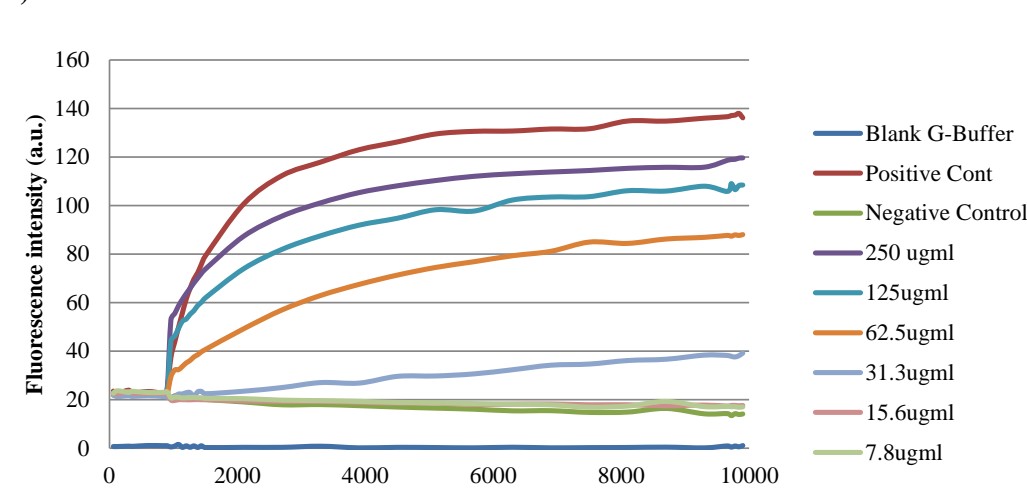

**(B)**

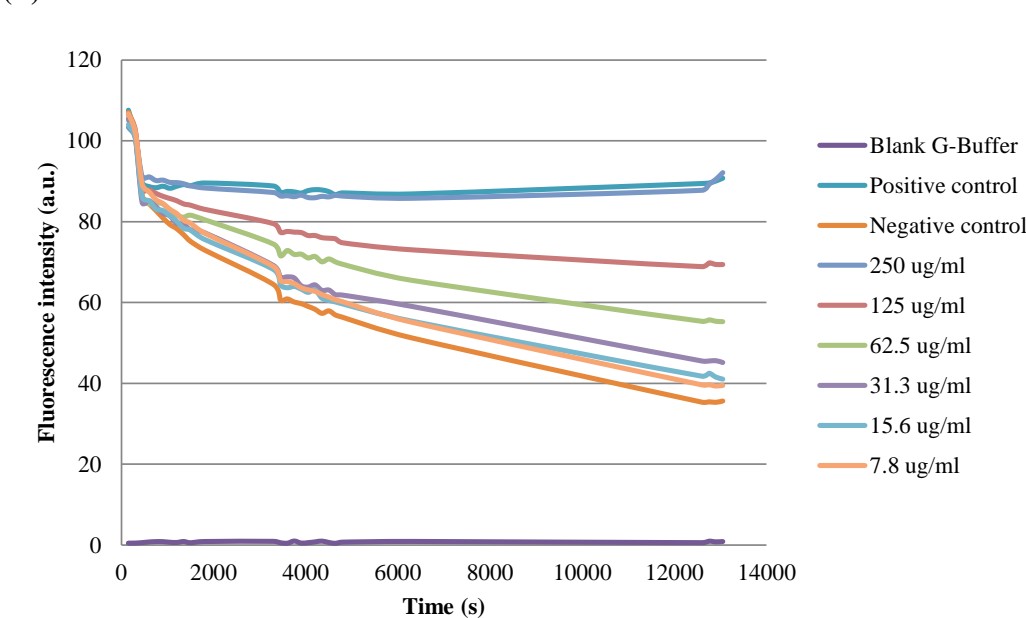

**Figure 6** **Effect of BipC-mediated nucleation of actin polymerization and depolymerization.** Samples containing 2 mM monomeric actin (10% Pyrene-actin) and fluorescence (expressed in Arbitrary Units, AU) were measured over time after initiation of polymerisation by His-BipC. (A) Actin polymerization of His-BipC at different concentrations and (B) actin depolymerization. Velocities were determined for the interval of 600–1200 s using the results of two independent experiments.

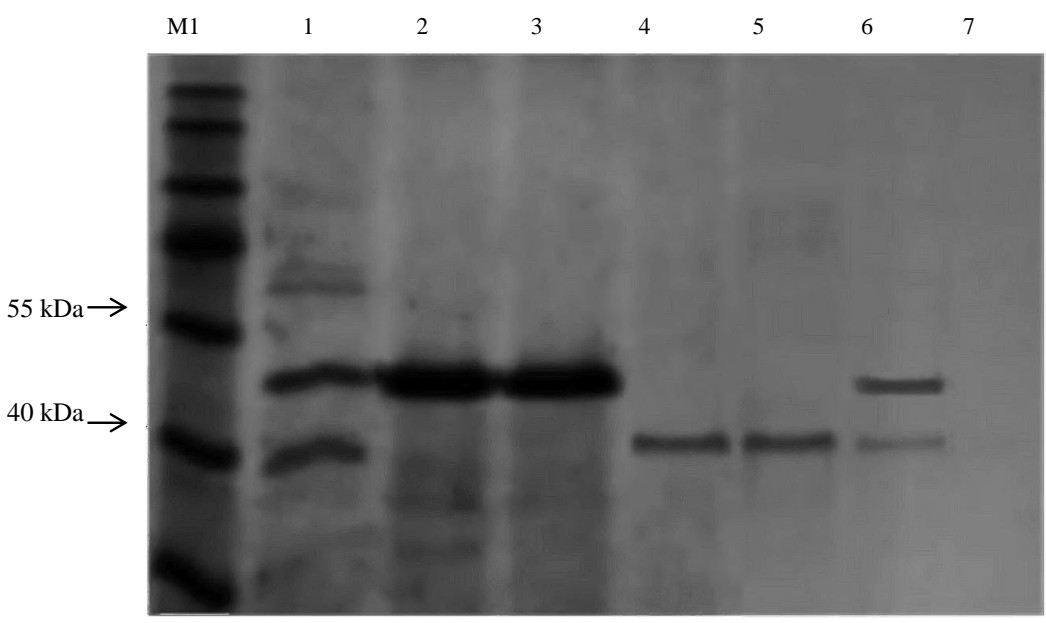

**Figure 7** *In vitro* **interaction analysis between His-BipC, F-actin, and G-actin by pull-down assays.** This protein pull-down assay demonstrated a possible interaction between BipC and both of the monomeric and filamentous actin. Lane M1, PageRuler prestained protein ladder; lane 1, His-BipC protein was immobilized on His-beads, followed by incubation with F-actin; lane 2, purified His-BipC; lane 3, purified His-BipC incubated with empty PBS buffer, lane 4, F-actin; lane 5, G-actin; lane 6, His-BipC protein with G-actin; lane 7, actin was immobilized on His-beads (control).

## Protein–protein interaction between BipC and actins

Protein–protein interaction was performed to confirm the binding of BipC with monomeric (G-actin), and filamentous (F-actin) actins. Both of the G- and F-actin were incubated with Dynabeads beads preloaded with His-BipC. The protein pull-down assay demonstrated an interaction between the BipC protein where the actin binding domain was able to associate with the filamentous actin. The His-BipC and F-actin interaction was obtained as shown in the SDS-PAGE (Fig. 7, lane 1). As compared to the monomeric actin, BipC appeared to bind primarily F-actin (Fig. 7, lane 6). This result suggests a role of BipC to promote actin bundling which is crucial in *B. pseudomallei* entry into host cell membrane.

## DISCUSSION

In this study, bioinformatics analyses were exploited in order to gain preliminary insights into the structural properties of BipC protein. According to *Sun & Gan (2010)*, BipC, which plays a role as a translocator protein may also play a role as an effector. Thus, the TTSEs bioinformatics tools were used to determine the role of BipC as an effector. The major features used for the TTSE prediction include the amino acid composition and positions, structural properties, and physiochemical properties (*Tay et al., 2010*; *Wang et al., 2013*). BipC harbor an N-terminal sequence that adequately fits the profile predicted by the secretion signal hypothesis. Presence of signal peptide encoded in the amino acid

sequence at the N-terminal of BipC protein by some of the TTSE tools, demonstrated that this protein as a pathogenicity island effector.

Interestingly, bioinformatics analysis of the entire BipC sequence identified a putative actin binding domain with sequence homology to the experimentally characterized IpaC_SipC family domain. Previous reports have indicated that IpaC and SipC harbors one actin binding domain (*Chatterjee et al., 2013*; *Knodler, Celli & Finlay, 2001*). SipC was determined as one of the effectors with proven actin modulatory domains that nucleates and bundles actin through its N- and C-terminal domains, respectively (*Hayward & Koronakis, 1999*). The amino acid residues 221–260 and 381–409 of SipC were shown to bind directly to and bundles F-actin (*Myeni & Zhou, 2010*). Besides that, the C-terminal region of SipC was also required for the secretion and translocation of effectors into host cells (*Chang, Chen & Zhou, 2005*). This dual functionality of SipC provides a strong evidence showing the functional complementation between effector domains (*Hayward & Koronakis, 1999*). In contrast, the SipC homologue IpaC from *Shigella* possesses the C-terminal actin nucleation domain (345–363), but with the absences of the N-terminal bundling domain (*Terry et al., 2008*). However, the N-terminus of this protein harbors sequences for the TTSS export and interaction with IpaB and IpgC. The central hydrophobic region is mainly involved in IpaB binding, invasion, and protein stabilization (*Kueltzo et al., 2003*; *Picking et al., 2001*). The predicted result of ordered/disordered regions of BipC showed the amino acid sequence 140–240 of BipC falls into both the IpaC_SipC superfamily domain region and the ordered region, and therefore could be used for further structural study.

Recombinant *bipC* was expressed and purified to further understand the functional roles of this protein at the molecular level. Prior to purification through SEC, there were multiple peaks observed on the SDS-PAGE of the Ni-NTA purified BipC. Size exclusion was then performed and the excess molecules were successfully removed from the Ni-NTA purified BipC. As a result, the purified BipC protein with a high purity of approximately 95% was obtained through the SDS-PAGE and HPLC analysis. Subsequently, CD spectropolarimetry was performed for the rapid determination of the secondary structure of BipC and to ensure that this protein was folded correctly. Samples for CD spectroscopy must be at least 95% pure by the criteria of HPLC, mass spectroscopy or gel electrophoresis. The CD spectra of the sample tested displayed a strong negative maxima around 197–200 nm region, suggesting that this protein possesses a well-defined alpha helix conformation. Notably, the CD spectra may change as a function of temperature, thus, leading to shifts in the baseline (*Miller et al., 1987*). Relative change due to influence of viable temperature was monitored in this study. A sharp transition from the native state to the denatured state was observed when the protein solution was gradually heated above a critical temperature. The transition midpoint was also determined from the graph and this indicated that the purified protein was folded. Coupled with the bioinformatics online tool, the secondary structure content of the molecule was comparable with the estimated alpha-helical conformation from the CD data. In addition, the structure of BipD has been crystallized and the overall structure consists of a bundle of antiparallel $\alpha$-helical segments with two small $\beta$-sheet regions (*Erskine et al., 2006*). Comparison of the structure of BipD, the predicted BipC structure in this study also possesses a well-defined alpha helix conformation.

The primary sequence of BipC contains a central alpha helix region and a C-terminal alanine-rich region. A comparison between the sequences of this protein demonstrated that BipC has sequence identity with SipC and IpaC at the carboxy-terminal region. This protein also harbors a distinct F-actin binding domain, explaining the ability of BipC to assemble actin filaments, as its homolog *Salmonella* SipC and *Shigella* IpaC. Tarp effector, a multifunctional protein, which primes the host cell for bacterial entry and survival intracellularly, was shown to harbor G-actin binding or nucleating domain and two distinct F-actin binding or bundling domains (*Jiwani et al., 2013*). These three domains mediate a direct link to the host cytoskeleton with the presence of discrete sites that are specifically associated with globular or filamentous actin. In contrast, F-actin-specific binding sites were also found in BipC, however, no typical actin-binding motifs such as the G-actin binding or nucleating domain is present in BipC. The absence of either of these actin-binding motifs raises the possibility of a novel molecular mechanism of actin assembly.

Previous studies on the bacterial effectors that influence host cell actin dynamics have provided valuable information to enhance the knowledge on the virulence factors targeting actin and contribute to a broader comprehension of actin dynamics in the eukaryotic cell (*McGhie, Hayward & Koronakis, 2001*; *Lee, Park & Park, 2014*). Thus, biophysical analysis was performed to further elucidate the functions of BipC protein in *B. pseudomallei* pathogenesis. In our previous knockout studies, BipC was shown to play a role in the actin-tail formation (*Kang et al., 2015*). The cell attachment, movement of cells, phagocytosis, intercellular replication, and the distribution of organelles are mostly depending on the presence of the actin (*Taylor, Koyuncu & Enquist, 2011*). In order to achieve these processes, bacteria secrete and inject effectors to hijack the host cell machinery. The actin cytoskeleton is one of the main targets of bacterial proteins and plays a key role in the host-pathogen interaction. Hence, the interaction and binding specificity of BipC especially with actin was further investigated via protein–protein interaction and functional *in vitro* assays.

In the *in vitro* actin polymerization and protein–protein interaction studies, BipC was shown to bind independently to the F-actin. This is in agreement with previous studies that have proven the contact-dependent secretion systems such as, TTSS to be involved in the actin binding activities in various other intracellular pathogens including *S. flexneri* and *S. typhimurium* (*Costa et al., 2015*). Previous studies have shown that the N-terminus of *Salmonella* SipC protein mediates F-actin bundling activity while the C-terminal region facilitates the actin nucleation activity (*Hayward & Koronakis, 1999*; *Chang, Chen & Zhou, 2005*; *Chang et al., 2007*). By analogy with the SipC protein, there is a high possibility of the BipC to have a similar actin nucleation and F-actin bundling activity.

In addition, BipC was found to bind F-actin and inhibit actin depolymerization *in vitro* without the aid of a bacterial or eukaryotic factor. Contrary to BipC, the pathogenic bacterium *Salmonella* SipC interacts with cellular actin and modulates its dynamics with the presence of the SipA (*Hayward & Koronakis, 1999*; *Zhou, Mooseker & Galan, 1999*). A study by *McGhie, Hayward & Koronakis (2001)* determined that SipC-SipA collaboration generates stable networks of F-actin bundle. Unlike *Salmonella* SipC, binding of F-actin by the BipC results in identifiable consequences whereby it could stabilize F-actin by blocking spontaneous depolymerization in order to enhance the bundling ability of the host cell

actin-binding protein. Thus, it is tempting to speculate that the ability of BipC in actin bundling may differ from the other intracellular pathogens.

The stable networks of F-actin bundles generated by BipC enhance the modulations of the eukaryotic cell cytoskeleton which is essential for pathogen internalization. Actin binding proteins (ABPs) present in the mammalian cells plays an important role in the remodeling of cellular actin filament network, which lead to the nucleation of actin polymerization or destabilization of F-actin to endorse the spatial control of filament assembly or disassembly (*Ayscough, 1998*). Furthermore, the cellular ABPs bundle has the ability to stabilize F-actin in order to generate higher order supramolecular structures that sustain membrane deformations (*Chen, Kaniga & Galan, 1996*; *Persson et al., 1997*; *Tran Van Nhieu, Ben-Ze'ev & Sansonetti, 1997*). Although BipC share no primary sequence similarity to the known eukaryotic ABPs, this protein was shown to bind directly to the actin and independently influences filament dynamics. Hence, this could be the 'add on' benefit for BipC as compared to other effector proteins.

Coupled with the biophysical study, our finding indicated that BipC may play a role in the host actin dynamic. The conserved ordered region could interact with or aid in the rearrangement of highly organized cortical actin polymer networks. BipC may be able to interfere with the host cell cytoskeletal network and recruit clathrin to the site of pseudopodia engulfment in order to assist *B. pseudomallei* in the cell-to-cell entry as reported in the study on *S. flexneri* (*Menard, Dehio & Sansonetti, 1996*). The ability of BipC to bind filamentous actin was associated with the presence of the actin-rich regions that play a role in modulating the organelle trafficking. Overall, our findings provide a crucial insight into a novel activity of BipC protein as an effector involved in the actin binding for the internalization of *B. pseudomallei* into the host cell. The architecture of BipC-mediated actin bundles has not been studied, and hence, it will be intriguing to analyze the structure of actin bundles formed for the purpose of host-cell infection in the near future.

## ACKNOWLEDGEMENTS

We are very grateful to Eunice Goh Tze Leng and Nandhu Muruganandham from Singapore Eyes Research Institute (SERI) for their technical assistance in the CD handling and biophysical study.

### Funding

This study was supported by Ministry of Higher Education (MOHE), Malaysia under the High Impact Research (HIR)-MOHE project (E000013-20001), University of Malaya Postgraduate Research Grant (PPP; PG161-2015B), and University of Malaya Research Grant (UMRG) (RP013C-13HTM). The funders had no role in study design, data collection and analysis, decision to publish, or preparation of the manuscript.

## Grant Disclosures

The following grant information was disclosed by the authors:

Ministry of Higher Education (MOHE), High Impact Research (HIR)-MOHE project: E000013-20001.

University of Malaya Postgraduate Research Grant: PPP, PG161-2015B.

University of Malaya Research Grant (UMRG): RP013C-13HTM.

## Competing Interests

The authors declare there are no competing interests.

## Author Contributions

- Wen Tyng Kang and Kumutha Malar Vellasamy conceived and designed the experiments, performed the experiments, analyzed the data, wrote the paper, prepared figures and/or tables, reviewed drafts of the paper.
- Lakshminarayanan Rajamani conceived and designed the experiments, analyzed the data, contributed reagents/materials/analysis tools, wrote the paper, reviewed drafts of the paper.
- Roger W. Beuerman conceived and designed the experiments, analyzed the data, wrote the paper, reviewed drafts of the paper.
- Jamuna Vadivelu conceived and designed the experiments, contributed reagents/materials/analysis tools, wrote the paper, reviewed drafts of the paper.

## Data Availability

The raw data has been supplied as Supplemental Files.

## Supplemental Information

Supplemental information for this article can be found online at http://dx.doi.org/10.7717/peerj.2532#supplemental-information.

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
