# Peer review of "Burkholderia pseudomallei type III secreted protein BipC: role in actin modulation and translocation activities required for the bacterial intracellular lifecycle"

_PeerJ, doi:10.7717/peerj.2532_

## Round 0.1 · original submission · Major Revisions

While the topic under investigation is of interest and the paper is generally well written both reviewers have raised serious concerns about the interpretation of the data and the controls used in some of the experiments. This is especially true in relation to the data presented in Fig 8 (actin-BipC binding). It is very likely that a comprehensive revision of the manuscript will require additional experimental data to satisfy the points raised. If you feel that you can address these points convincingly I am willing to consider a revised manuscript.

Reviewer 1 ·

Basic reporting

The standard of English language is generally very good, but there are some grammatical errors.

The introduction clearly shows the context of the study. It could be improved by also including information on the role of BipC in actin polymerisation and cytotoxicity as assessed with BipC mutant bacteria in JID 2015, 211:827, and information on the evidence as to why BipC is predicted to be a translocator.

Figure 1 is unnecessary as the similiarity of BipC to SipC has previously been reported. Figure 2 could be deleted or placed as supplementary data. Figure 4A is also unnecessary. The presentation quality of Figure 5 should be improved.

The title of the manuscript should be altered to more appropriately reflect the content of the study, as intracellular trafficking was not directly investigated.

Experimental design

The research study is original and within the scope of PeerJ.

The research question is well defined, relevant & meaningful and aims to increase understanding of the role, and mechanism of action, of TTSS and specifically BipC in Burkholderia pseudomallei infection.

The methods are described with sufficient detail & information to replicate.

Validity of the findings

In this study Kang et al purified the Burkholderia pseudomallei TTSS protein BipC and investigated its ability to interact with actin and to affect actin polymerisation. The study seems too preliminary for publication at the current time because several experiments are missing appropriate controls and some conclusions are not fully supported by the data.

Table 1. The presence of a TTSS signal peptide does not predict that BipC is an effector as a similar signal peptide would be present on a translocator.

The authors have successfully expressed His-BipC in E. coli and purified it. Fig 4B could be improved by showing samples from each stage of the purification process.

Fig 6. BipC is predicted to be a TTSS effector that is delivered into host cells by the TTSS. How do the authors propose that pure protein enters the cells to cause cell death? Cells should also be co-incubated with another His-tagged protein as a negative control, to show that the cell death is specific caused by BipC and is not due a non-specific effect due to 24h exposure with a recombinant bacterial protein.

Fig 7. The data in Fig 7B do not seem to support the conclusions as the negative and positive controls give the same result (though perhaps the Figure legend has been mis-labeled?). If decreased amounts of depolymerised amount of actin are detected in the presence of BipC this could be due to BipC reducing depolymerisation as suggested by the authors. An alternative interpretation is that BipC is re-polymerising any actin that is depolymerised so that a steady level of polymerised actin is detected.

Fig 8. To show that the actin binding is specific for BipC, another His-tagged protein should be used as a negative control in the pull-down assay. In addition the following controls are needed in the pull-down assay: G-actin alone, F-actin alone and BipC alone. The amount of BipC recovered is very low in comparison to actin - how much of each protein was added to the assay and what was the molar ratio? Polymerization of actin is not shown in lane 4 as the size of monomeric actin is 42kD.

Additional comments

In this study Kang et al purified the Burkholderia pseudomallei TTSS protein BipC and investigated its ability to interact with actin and to affect actin polymerisation. The study seems too preliminary for publication at the current time because several experiments are missing appropriate controls and some conclusions are not fully supported by the data.

Reviewer 2 ·

Basic reporting

Previous work by the authors described the deletion mutant of the BPS T3SS effector protein, BipC. Presently, they report the predicted structure of the same protein using various bioinformatics tools and characterise the purified protein. The report is a useful description of an important virulence factor of this pathogen, utilising standard techniques and making appropriate conclusions. They draw upon similarities and make distinctions between orthologous proteins in other species.
Specific points that would improve the discussion are suggested addressed below:

*The related protein, BipD has been crystalised. How do the predicted BipC structural features compare?
* The authors state that BipC did not show good homology with any other protein in the protein data bank. Could the authors comment on this given the expected similarity with SipC, or even BipD?
* In the author’s previous work, the BipC mutant improved survival of host cells by about 30%, but there was still approximately 30% cell death. Could the authors comment on other proteins that may be responsible for cytotoxicity in addition to BipC?
* A key feature the authors identify is the ability of BipC to induce cytotoxicity. However, in addition they demonstrate that BipC has a role in actin polymerisation and invasion at similar/equivalent concentrations. How do the authors resolve these opposing effects, and what is the consequence for the infection process?

Experimental design

* How do the authors calculate a purity of 95%?* How do the authors calculate a purity of 95%?
*Does the size of the purified protein match the expected size of pure BipC?
* Was the PCR product used for cloning and expression sequenced to confirm BipC?
* Do the authors anticipate any influence of the HIS-tag on protein folding and function? Is there a control they could use to test this?

Validity of the findings

* Fig5B) contains two lines but it is not clear from the legend what each represents.
* Fig8) Several issues with the protein-protein interaction experiment need to be resolved:
How were proteins that were adsorbed to dynabeads removed prior to the SDS-PAGE? The band showing BipC binding is very weak, suggesting poor binding or elution. It is also missing from the Actin-BipC conditions, should there not be two bands, one for each protein, after elution of the proteins from the bead? What is the saturation capacity of the beads? How can the authors exclude direct binding of actin to the beads independently of BipC? Finally it is not clear how the authors conclude from Fig8 that BipC is able to polymerise G-actin. Related to this, they discuss that BipC contains no G-actin binding motifs, which seemingly contradicts their comments in the results section on protein-protein interaction.

Additional comments

The manuscript represents a useful addition to our knowledge in the field and in general there are only minor points to address. The only major concern is the interpretation of the data shown in Figure 8, which does not seem to support the conclusions made by the authors.

---

## Round 0.2 · accepted · Accept

After a careful consideration of your responses to the reviewers comments as well as the modifications you have made to the paper I am happy to accept this version for publication. Thank you for taking the time to address the reviewers suggestions and improve the manuscript.